# *Rag1* and *Rag2* Gene Expressions Identify Lymphopoietic Tissues in Larvae of Rice-Field Eel (*Monopterus albus*)

**DOI:** 10.3390/ijms23147546

**Published:** 2022-07-07

**Authors:** Yuchen Liu, Nan Jiang, Wenzhi Liu, Yong Zhou, Mingyang Xue, Qiwang Zhong, Zhong Li, Yuding Fan

**Affiliations:** 1Yangtze River Fisheries Research Institute, Chinese Academy of Fishery Sciences, Wuhan 430223, China; liuyuchen0125@163.com (Y.L.); jn851027@yfi.ac.cn (N.J.); liuwenzhialisa@yfi.ac.cn (W.L.); zhouy@yfi.ac.cn (Y.Z.); xmy@yfi.ac.cn (M.X.); 2National Demonstration Center for Experimental Fisheries Science Education, Shanghai Ocean University, Shanghai 201306, China; 3College of Biological Science and Engineering, Jiangxi Agricultural University, Nanchang 330045, China; zhongqiwang@jxau.edu.cn

**Keywords:** rice-field eel, *Monopterus albus*, lymphopoiesis tissues, *rag1*, *rag2*

## Abstract

In immature lymphocytes, recombination activating genes 1 and 2 are necessary for antigen receptor V (D) J recombination, representing immature lymphocyte biomarkers. Herein, we cloned and sequenced rice-field eel *rag1* and *rag2* genes. Their expressions in the thymus, liver, and kidney were significant from 0 days post hatching (dph) to 45 dph, peaking at 45 dph in these three tissues. In situ hybridization detected high *rag1* and *rag2* expressions in the liver, kidney, and thymus of rice-field eel from 2 to 45 dph, suggesting that multiple tissues of rice-field eel contain lymphocyte lineage cells and undergo lymphopoiesis. Tissue morphology was used to observe lymphopoiesis development in these three tissues. The thymus primordium began to develop at 2 dph, while the kidney and liver have generated. Our findings verified that the thymus is the primary lymphopoietic tissue and suggested that, in rice-field eel, lymphocyte differentiation also occurs in the liver and kidney.

## 1. Introduction

The rice-field eel (*Monopterus albus*), belonging to the family *Synbranchidae*, is an important economic fish species in China [1]. Deterioration of the water environment and overfishing have caused the wild population to decrease sharply [2]. In addition, because of the increasing demand for rice-field eel, artificial breeding commenced countrywide in 1950 [3]. Unfortunately, artificial high-density breeding resulted in various diseases, which caused high mortality, with detrimental effects on the rice-field eel breeding industry. Thus, disease control has become a key element to the success of rice-field eel artificial breeding. To effectively prevent, control, and treat the diseases and improve the survival rate, the immune system of rice-field eel should be studied. The rice-field eel, which is different from ordinary fish, has special biological characteristics, for example, it displays a sex reversal phenomenon [4]. Previous studies on rice-field eel mostly focused on embryonic and post embryonic development [5] and gonadal development [6]; however, there has been a lack of research on the developmental biology of the immune system.

Animals rely on the function of the immune system to resist diseases and maintain health. The immune system comprises immune organs, cells, and active substances. In cooperation with other body systems, the immune system identifies and eliminates antigenic foreign entities, thereby sustaining the internal environment in a stable and physiologically balanced state [7]. The immune system has evolved during speciation. Invertebrates only have a simple phagocytic function to defend against inflammatory effects. Fish have evolved immune organs, such as thymus and spleen, which involve in T and B lymphocyte development. Birds have a special immune organ, the bursa fabricius, which produces antibodies. Mammals have a relatively complete immune system, including innate and adaptive immune systems [8]. Compared with mammals, the rice-field eel, as a kind of bony fish, lacks bone marrow and lymph nodes [8] and its lymphopoietic tissues mainly comprise the thymus, kidney, and spleen [9]. Immune cells, such as lymphocytes, macrophages, and granulocytes, are distributed in liver, digestive tract, gills, skin, and mucosal lymphoid tissues, and these tissues also have immune effects. The developmental order of immune organs is different between freshwater fish and seawater fish. Although the earliest lymphoid organ in freshwater fish is the thymus, and the kidney and spleen is developed earlier than thymus in some seawater fish, the thymus is also the first organ to become lymphoid [10]. In rock bream [11] and Atlantic cod [12], the earliest immune organ becoming lymphoid is the kidney. However, because of the wide variety of fish, the immune systems of limited fish species have been studied, and we lack comprehensive research on the developmental differences of fish immune organs [13].

The recombination activating genes 1 and 2 (*rag1* and *rag2*) are DNA recombination activating genes, representing key genes of the vertebrate specific immune response [14]. *Rag* genes encode recombinase proteins that can promote V (D) J rearrangement so as to change the antibody genotype and produce a new specific immune response [15,16]. The V (D) J recombination dominated by *rag* genes results in a variety of immunoglobulins and T cell receptor subtypes, enriches specific immunity, and affects the development and maturation of lymphocytes and the immune system [17]. Ig is secreted by B lymphocytes, while TCR is expressed by T lymphocytes. The recombination signal sequence exists in the V (D) J region of Ig and TCR. *Rag1* can recognize and bind RSS alone, while *rag2* can bind to DNA only in the presence of *rag1*. In addition, rag1 and rag2 genes can form complexes, which have higher specificity and stability and can be recognized more effectively [18,19]. *Rag* genes, especially *rag1*, are widely used in the study of phylogeny because of their slow evolution and highly conserved sequences among different species [20]. In the study of lymphocyte development in lower vertebrates, the *Xenopus* laevis thymus was observed as the main site of *rag1* and *rag2* expressions, and both genes were expressed in the thymus and bone marrow of adult frogs of all ages [21]. In the axolotl thymus, *rag1* was continuously expressed until the thymus degenerated naturally after 12 months, with peak expression occurring in the thymus at 4.5 months [22]. The expression of *rag1* in rainbow trout was limited to immunoglobulin negative lymphocytes on the surface of thymus, while *rag2* was expressed mainly in anterior kidney and thymus [16,23]. In zebrafish, *rag1* was expressed in thymus and kidney at 4 dpf (days post fertilization), while *rag2* was mainly expressed in the thymus [20]. *Rag1* of grass carp was expressed in the thymus at 4 dpf and *rag2* expressed at 7 dpf, while the expressions of both genes persisted in the thymus during the first age [24]. Thus, previous research suggested that lymphocyte development varied among fishes. There have been no reports of *rag1* and *rag2* genes in the lymphopoietic tissues of rice-field eel. Thus, *rag1* and *rag2* temporal and spatial expression patterns could be used to reveal the generation of lymphocyte in the rice-field eel.

In this study, in situ hybridization and quantitative real-time reverse transcription PCR (qRT-PCR) were used to investigate *rag1* and *rag2* temporal and spatial expression patterns in different development stages. In addition, hematoxylin and eosin (HE) staining was used to analyze morphological and structural characteristics and the changes in lymphopoiesis tissues during development.

## 2. Results

### 2.1. Evolutionary Tree Analysis

The phylogenetic tree analysis of RAG1 and RAG2 showed that rice-field eel RAG1 and RAG2 clustered with the *Paralichthyidae* and were closely related to the proteins from rainbow trout and zebrafish (Figure 1 and Figure 2).

### 2.2. Rag1 and Rag2 mRNA Expressions

According to the qRT-PCR results, the expression of rag1 and rag2 was detected in the kidney, liver, and thymus from 0 to 45 dph in rice-field eel (Figure 3A,B). In the early development stage of the thymus, the expression level of rag1 was high, from 0 dph to 17 dph, and maintained at a low but significant level from 24 dph to 30 dph. At 45 dph, the expression level was the highest. Meanwhile, the expression level of rag2 in the thymus was low from 0 to 30 dph, and then a high expression level was detected at 45 dph. In the liver and kidney, the expression levels of both rag1 and rag2 were high from 0 to 6 dph and then maintained at low but significant levels, and the highest expression levels were detected at 45 dph.

### 2.3. In Situ Hybridization (ISH) Detection of Rag1 and Rag2 Transcriptions

The qRT-PCR results were verified using ISH detection of rag1 and rag2 expression profiles. ISH was used to analyze rag1 and rag2 expressions in the kidney, liver, and thymus to determine the temporal and spatial localization of immature lymphocytes in various organs and developmental stages. Rag1 and rag2 expression signals were detected at the pharyngeal epithelium at 2 dph (Figure 4A,E). After 10 dph, rag1 and rag2 transcripts were observed at the thymic primordium periphery (Figure 4B,F). After 24 dph, the rag1 and rag2 signals were scattered randomly throughout the whole thymus (cortex and medulla) (Figure 4C,G). After 45 dph, the rag1 and rag2 signals were observed mainly in the thymus sub-capsular region, with lower signals in the medulla (Figure 4D,H). From 2 to 45 dph, the rag1 and rag2 signals were distributed randomly throughout the whole liver (Figure 5). In the kidney, the rags signals were detected in the tubular epithelia at 2 dph. From 6 to 45 dph, the rag1 and rag2 signals were observed in the hematopoietic tissues of kidney (Figure 6). No rags signal was detected when labeled with sense probes used as negative control (Appendix A).

### 2.4. Histological Analysis of the Genesis of Lymphopoiesis Tissues

As a complement to the ISH analysis, morphological changes to the thymus, kidney, and liver of rice-field eels from 2 to 45 dph were examined by histology. The thymic primordia of the rice-field eel were formed by 2 dph and were distributed on the pharyngeal and laryngeal epithelia (Figure 7A). The thymus formed a loose reticular structure after 6 dph, in which lymphocytes gathered and began to differentiate (Figure 7B). Subsequently, the size of thymus and the number lymphocytes increased. After 10 dph, connective tissue and capillaries began to extend into the parenchyma of the thymus, and its tissue structure became denser (Figure 7C). At 45 dph, the thymus of rice-field eel showed zoning, with sparse cell distribution and shallow staining in the inner area. There were lymphocytes in the outer area, which were small and dense, and the cells were stained deeply (Figure 7F).

The head kidney tissue appeared at 2 dph (Figure 8A). There were several curly renal tubular structures, which were roughly divided into left and right lobes, the base of which was connected. A small number of blood cells and lymphocytes could be seen around the renal tubules. The number of blood cells and lymphocytes around the renal duct increased after 6 dph (Figure 8B). The blood cells around the renal tube formed a mass and the renal tube expanded after 10 dph (Figure 8C). After 17 dph, the volume of head kidney increased, lymphocytes gathered, and blood cells were still distributed around the renal tube (Figure 8D). After 24 dph, the renal ducts were surrounded by lymphocytes and granulocytes (Figure 8E). After 45 dph, the renal tube of rice-field eels began to degenerate, the epithelial cell became blurred, and the amount of hematopoietic tissue increased (Figure 8F).

The liver has generated at 2 dph, with significant numbers of hepatic sinusoids (Figure 9A). With the continuous development of the fish body, the liver volume increased. The hepatocyte cord became dense, and the hepatic sinus narrowed. After 24 dph, the liver lobules of rice-field eels were not obviously divided, and the liver parenchyma was dense, with less connective tissue structure (Figure 9D).

## 3. Discussion

*Rag1* and *rag2* genes encode recombinant activator proteins, which mediate B and T cell receptor recombination during the development of lymphocytes. Therefore, their expressions are useful and reliable markers of lymphocyte differentiation in lymphopoietic tissue [25]. Herein, we observed that the rice-field eel RAG1 and RAG2 proteins were phylogenetically closely related to their homologs in *Paralichthys olivaceus*. *Rag1* and *rag2* genes expression of the liver and kidney in rice-field eel were similar, starting as early as 0 dph and peaking at 45 dph. However, in the thymus of rice-field eel, the expression level of *rag1* was high from 0 dph to 17 dph and peaking at 45 dph, while *rag2* expression maintained at a low level from 0 to 30 dph, peaking at 45 dph. In fish, the thymus and kidney were sites for lymphocytes differentiation [22,26]. In zebrafish, *rag1* was expressed in thymus and kidney at 4 dpf (days post fertilization), while *rag2* was mainly expressed in the thymus [20]. *Rag1* of grass carp was expressed in the thymus at 4 dpf and *rag2* expressed at 7 dpf, while the expressions of both genes persisted in the thymus during the first age [24]. The thymus is an important immune organ. Previous studies showed the relevant characteristics of the thymus of teleost fish, for example, the main lymphoid and hematopoietic tissue development of the zebrafish [27] and the origins of T and B lymphopoiesis in agnatha and jawed fish [28]. Similar to other bony fish, the rice-field eel *rag1* and *rag2* are mainly expressed in the thymus and kidney, as expected. Previous researchers found that, in the early development stage of *Xenopus*, *rag1*, and *rag2*, expression was mainly observed in the thymus, which was the site for T cell differentiation [21]. *Rag1* gene expressed at the surface of thymus and *rag2* gene expressed in both thymus and head kidney, which indicated that thymus and head kidney was primitive lymphoid tissue in rainbow trout [16,23]. *Rag1* and *rag2* genes were expressed in the head and body kidneys of *Paralichthys olivaceus*; therefore, the kidney was the lymphoid organ of *Paralichthys olivaceus* [29]. Moreover, rag gene activity was also detected in the liver and spleen, which was necessary for B cell differentiation of *Xenopus* [30]. In Chinese giant salamander and Mexican axolotl, the liver was considered as a site for B cell differentiation [22,31]. The activity of *rags* induces V (D) J rearrangement during B cells development. In B cell development, *rags* transcriptions levels were higher in pro-B cells and pre-B cells and lower in lymphoid stem cells, immature B cells, naive B cells, and mature B cells [32]. In this study, the presence of two peaks of *rags* transcriptions might indicate the presence of pro-B cells and pre-B cells in the liver and kidney. Therefore, the liver and kidney were considered to be lymphopoiesis tissue. In addition, by analyzing the expression patterns of *rag1* and *rag2* in the kidney, liver, and thymus at different development stages, the development of these lymphopoiesis tissues in rice-field eel could be inferred.

The thymus of teleosts is a paired organ, which is generally located at the back of the gill cavity and the upper dorsal horn at the junction of the operculum and the pharyngeal cavity. The thymus is covered by the operculum mucosa and is distributed symmetrically [33]. The location of the thymus in the rice-field eel is similar to that of most fish. The development of the thymus begins with the thymic primordium, the stem cells migrate into small lymphocytes, and the circulating lymphoblasts colonize under the pharyngeal epithelium. Then, the thymus begins to develop into a lymphoid organ, and the blood vessels and connective tissue in the thymic parenchyma increase, as do the reticular fibers. Afterwards, the interstitial components of thymus are gradually enriched, and the thymus is divided into sections, ultimately forming a lobulate thymus [34]. The development time points of lymphocytes and the development of thymus structure are different among species. The thymus primordium of zebrafish appeared at 54 h after fertilization and the *rag* genes expressed at 4 dph, which suggested that the thymus appeared earlier than the differentiation of lymphocytes [35]. The thymic primordia of carp appeared about 2 dph [36], and that of sea bream appeared at 11 dph [37], while, that of *Paralichthys olivaceus* appeared at 10–15 dph [38], and the thymic primordia of Atlantic sturgeon appeared at 48 dph [39]. The thymus of cobia began to appear at 7 dph, and lymphocytes appeared in the thymus at 15 dph, and the formation of immune memory was consistent with the formation of the thymus [40]. In the present study, the thymic primordium was observed at 2 dph. *Rags* signals positive immature lymphocytes were detected at the edge of the thymus at 10 dph and randomly on the surface and inside the thymus at 24 dph. At 45 dph, immature lymphocytes were intensively expressed in the outer thymus; meanwhile, the thymus of the rice-field eel showed internal and external divisions, and the mature thymus generated. However, the thymus of the rice-field eel did not show lobulation, which was different from that of *Epinephelus obliquus* [41] and Chinese sturgeon [42]. Researchers found that the thymus of *Xenopus* showed the division of the cortex and the medulla in the early stage of embryonic development [43], while in this experiment, the division of the thymus cortex and medulla occurred at 45 dph.

The kidney of teleost fish is usually divided into head, middle, and posterior kidneys. The middle kidney and posterior kidney are not only urinary organs but also hematopoietic tissue, which function urination, regulation of osmotic pressure, and hemopoiesis [44]. However, the structure of the head kidney is relatively simple, which gradually loses its excretory function in the process of differentiation and becomes a hematopoietic, immune, and endocrine organ [45]. *Rag1* and *rag2* signals were observed in the tubular epithelia at 2 dph and then distributed randomly in the hematopoietic tissues of kidney from 6 to 45 dph. It seems that the renal tubules might be transient lymphopoietic tissues before hematopoietic tissues generation in the rice-field eel. The expression of *rags* genes in the kidney varied among fish species. The expression of *rag1* in rainbow trout was low in the head kidney [23], while in zebrafish, *rag1* expressed in the head kidney [46]. *Rag1* in carp expressed in the kidney at 7 dph but was not detected in the kidney of 1-year-old carp, which proved that the head kidney of carp had a degraded immune function [47]. The kidney of *Paralichthys olivaceus* appears at the time of hatching stage [38]. The head kidney primordia of orange-spotted grouper appeared at 10 dph, and the renal tubules began to degenerate at 50 dph [41]. The development of large yellow croaker was slower, the head kidney tissue appeared at 3 dph, and the renal tubules began to degenerate at 60 dph; moreover, the head kidney was not completely transformed into lymphoid organs until 10 months old [48]. Zhong et al. found that the kidney of catfish developed faster, it appeared on the hatching stage, and the renal tubules disappeared completely at 28 dph [49]. The head kidney tissue of the rice-field eel appeared at 2 dph, the head kidney tissue appeared, and several renal tubules showed curl distribution. At 17 dph, the volume of head kidney increased, lymphocytes gathered, and blood cells were still distributed around the renal tube. Then, the renal tube degenerated after 45 dph. In addition, lymphocytes appeared in the head kidney of rice-field eel at an earlier time than that in the thymus, while in *Paralichthys olivaceus* and golden head snapper, lymphocytes appeared earlier in the thymus and lymphocytes migrate from the thymus to the head kidney [36,50].

The liver is one of the important digestive glands of teleost fish. It has the physiological functions of material metabolism, detoxification, hematopoiesis, and excretion. In addition, it is also the organ that can best reflect the physiological, pathological, and nutritional status of the body [51]. In the axolotl, the liver is considered to be a transient lymphoid tissue, and the *rag1* signals could be detected in the hematopoietic layer [22]. The liver of rice-field eel also showed *rag1* and *rag2* signals distributed in the entire tissues, indicating that the liver could be regarded as lymphopoietic tissues that function in the immune response. The livers of different fish have different development patterns. The liver rudiment of *Epinephelus obliquus* appeared about 3 dph, while liver cells formed hepatocyte clusters [52]. The liver of loach develops into a mature structure at 11 dph [53]. The rice-field eel liver development was similar to that of the big yellow croaker. At 2 dph, liver cell clusters both appeared in the rice-field eel and the big yellow croaker; meanwhile hepatocytes differentiated, and fat was stored in them. Subsequently, hepatocytes grew, and the volume of the liver increased [48]. The liver lobes of the rice-field eel were not obvious, while the liver of the big yellow croake was divided into two lobes at 10 dph. There are three arrangements of hepatocytes in the liver, one of which, such as in *Culter Alburnus*, takes the central vein as the center, and the hepatocytes are arranged radially to form mutually consistent hepatocyte cords, and hepatic blood sinuses are formed between adjacent hepatocytes [54]. The results of the present study showed that the liver of rice-field eel was also arranged in this way.

Juvenile fish are very fragile and easily affected by external environmental stimuli, thus their immune function is particularly important. It is generally believed that vertebrates rely on the lymphoid tissues and organs to produce immune activity, and the functional maturity of lymphocytes always occurs later than the development of lymphoid tissues and organs [55]. Therefore, the development of lymphopoietic tissue might not be completed until the juvenile stage, and its immune function might mature later. Our findings showed that the thymus primordia in rice-field eel appeared at 2 dph; however, the morphological changes lasted 1.5 month. Therefore, the juvenile rice-field eels are more prone to disease outbreak may because of their immature immune function. We also noticed that the thymus primordium began to develop while the kidney and liver generated. Accordingly, in the rice-field eel, the liver and the kidney developed earlier than the thymus, and the kidney was the first immune organ becoming lymphoid, which is similar to the development order of rock bream [11] and Atlantic cod [12]. In rice-field eel larvae, the spleen was too small to dissect cleanly; therefore, it was not analyzed in this study.

In conclusion, rice-field eel RAG1 and RAG2 are closely related to their homologs in *Paralichthys olivaceus*. High *rag1* and *rag2* expression levels were used as a proxy to detect lymphocyte differentiation from 0 dph. The *rag1* and *rag2* spatial expression patterns suggested that, in addition to the thymus, the liver and kidney are also lymphopoietic tissues, which support lymphocyte receptor rearrangement in the rice-field eel.

## 4. Materials and Methods

### 4.1. Animals

The Animal Care and Use Committee of the Yangtze River Fisheries Research Institute, Chinese Academy of Fishery Sciences, approved all the experimental procedures and animal handling.

An experimental breeding base in Xiantao City, Hubei Province, provided the rice-field eels, which were all members of the same spawning batch. Fish were maintained in the laboratory at 25 °C. They were distributed in a 30 × 100 cm boxes with shading, and the water was changed once a day. The aerator was used to provide the oxygen needed by the rice-field eel. Fish were fed with red worm once a day. The tissues (thymus, kidney, and liver) were sampled at 0, 2, 6, 10, 17, 24, 30, and 45 days post hatching (dph). In total, 120 fish that were spawned at one time were used as an experiment. Every 40 fish were kept in the same tank, for a total of 3 tanks. At total of 5 fish were taken from each tank at each time point, and a total of 15 fish were used as a sample. We used three different batches of fish as biological replicates. The experiments were performed in triplicate as technical replicates. The experiments were performed in triplicate.

### 4.2. Extraction of RNA and Synthesis of cDNA

The TRIzol reagent (Invitrogen, Waltham, MA, USA) was used to extract total RNA from the designated tissues, following the supplier’s instructions, which was stored at −80 °C before use. A cDNA Synthesis SuperMix kit (TransGen Biotech, Beijing, China) was used to reverse transcribe the RNA into cDNA, following the supplier’s instructions.

### 4.3. Phylogenetic Tree Analysis

The full length RAG1 and RAG2 proteins of rice-field eel were downloaded from the NCBI database, and their accession numbers were XP020470881 and XP020470878, respectively. The RAG1 and RAG2 protein sequences of various organisms were also downloaded from the NCBI database for homology analysis, and a multiple alignment of the amino acid sequences was carried out using the Clustal W 2.0.10 software (European Bioinformatics Institute, Cambridge, UK) [56]. Phylogenetic tree analysis was then carried out using the maximum likelihood (ML) method in the Mega 7.0 software [57], with 1000 bootstraps tests.

### 4.4. qRT-PCR Analysis

Following reverse transcription of mRNA from various tissues, the resultant cDNA was used as the template for the quantitative real-time PCR step of the qRT-PCR protocol using designed primers (Table 1). The reaction comprised: 2 µL of cDNA template, 0.4 µL of 0.2 µM forward and reverse primers, 10 µL of Hieff qPCR SYBR Green Master Mix (Yeasen, Shanghai, China), and diethylpyrocarbonate (DEPC) water to a total volume of 20 µL. The qPCR cycling conditions comprised: 95 °C for 5 min; 40 cycles of 95 °C for 10 s, 60 °C for 20 s, and 72 °C for 20 s; followed by a melting stage at 95 °C for 15 s, during which the temperature was increased by 0.2 °C/s from 60 to 95 °C. Each sample was run in triplicate.

### 4.5. In Situ Hybridization

A digoxygenin (DIG) RNA-labeling kit (Promega, Madison, WI, USA) used SP6 RNA polymerase to synthesize the *rag1* and *rag2* probes, following the supplier’s instructions. Table 1 shows the primers used to prepare the probes. The negative controls comprised the sense probes for each gene. In situ hybridization on sections was carried out following a previously described protocol [31,47]. Tissues were fixed using 4% paraformaldehyde, dehydrated using 30% sucrose/phosphate-buffered saline (PBS,) and embedded in optimum cutting temperature compound (OCT, Leica, Wetzlar, Germany). For each tissue, we acquired 9 µm transverse serial sections at −20 °C (UC7, Leica). Three incubations with PBS were used to rehydrate the sections, followed by incubation with 10 µg/mL protease K for 20 min. The sections were then treated with pre-hybridizing solution, added with the digoxygenin (DIG)-labeled probes, and incubated at 65 °C overnight. Subsequently, formamide and SSC were used to treat the sections, which were then reacted with anti-DIG antibody (1:4000; Roche Diagnostics GmbH, Basel, Switzerland). Nitro-blue tetrazolium, 5-bromo-4-chloro-3′-indolyphosphate (NBT/BCIP; Roche Diagnostics), and 3, 3′-diaminobenzidine (DAB; Roche Diagnostics) were then used to stain the sections, followed by observation under a microscope (DM2500, Leica).

### 4.6. HE Staining and Histology

We used PBS to rehydrate the transverse serial sections of the kidney, liver, and thymus, followed by HE staining. Thereafter, the sections were dehydrated using an ethyl alcohol gradient, cleared using xylene, sealed using neutralized resin, and examined under light microscopy.

### 4.7. Statistical Analysis

The results for the three replicates of the qRT-PCR analysis of each sample were expressed as the mean (±the standard deviation (SD)). According to our previously published paper, *ef1*α was selected as the housekeeping gene [58]. In each sample, each gene’s mRNA level was calculated as the ratio to the mRNA level of *ef1*α.

## Figures and Tables

**Figure 1 ijms-23-07546-f001:**
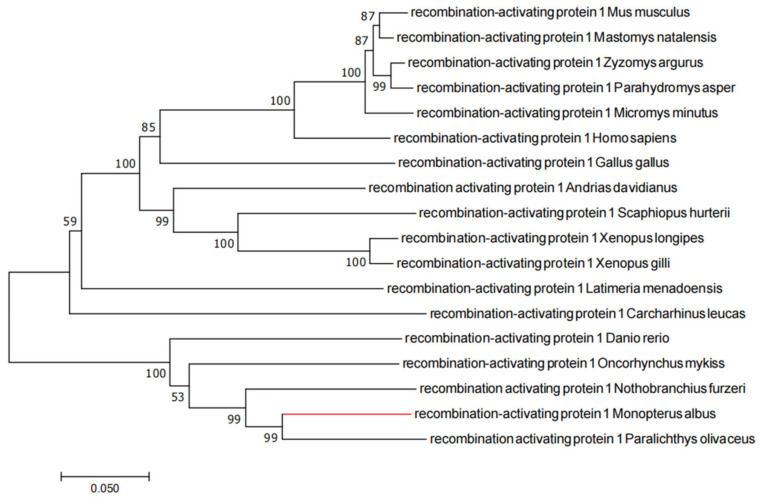
Phylogenetic tree of RAG1 proteins. The phylogenetic tree was constructed based on the full-length amino acid sequences by the maximum likelihood (ML) method, and the bootstrap value was set at 1000. GenBank accession numbers: *Mus musculus* (NP_033045); *Mastomys natalensis* (ACA47349); *Zyzomys argurus* (ACA47369); *Parahydromys asper* (ACA47358); *Micromys minutus* (ACA47353); *Homo sapiens* (NP_000439); *Gallus* gallus (NP_001026359); *Andrias davidianus* (AWS00967); *Scaphiopus hurterii* (ABS00331); *Xenopus longipes* (ABS00359); *Xenopus gilli* (ABS00345); *Latimeria menadoensis* (AAS75807); *Carcharhinus leucas* (AAB17267); *Danio rerio* (NP_571464); *Oncorhynchus mykiss* (NP_001118209); *Nothobranchius furzeri* (KAF7224726); *Monopterus albus* (XP_020470881); *Paralichthys olivaceus* (AIK29461).

**Figure 2 ijms-23-07546-f002:**
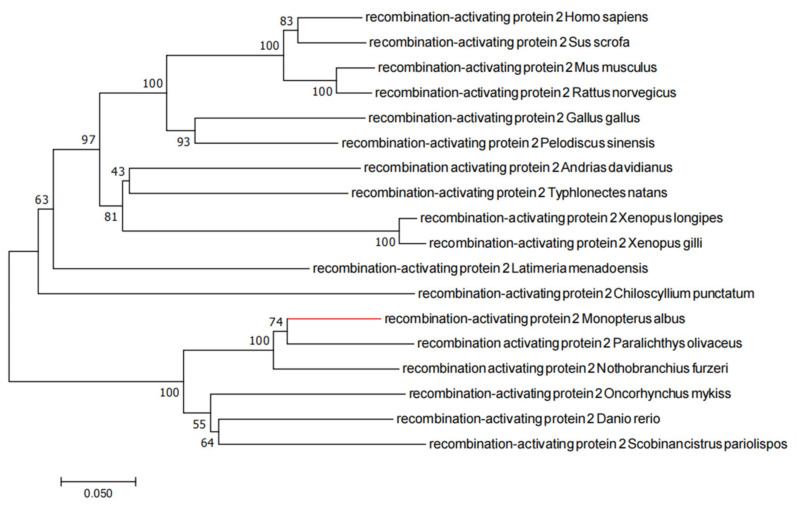
Phylogenetic tree of RAG2 proteins. The phylogenetic tree was constructed based on the full-length amino acid sequences by the maximum likelihood (ML) method, and the bootstrap value was set at 1000. GenBank accession numbers: *Homo sapiens* (NP_000527); *Sus scrofa* (NP_001121953); *Mus musculus* (NP_033046); *Rattus norvegicus* (NP_001093998); *Gallus gallus* (NP_001291986); *Pelodiscus sinensis* (AAL12891); *Andrias davidianus* (AWS00968); *Typhlonectes natans* (AAL12890); *Xenopus longipes* (ABS00392); *Xenopus gilli* (ABS00369); *Latimeria menadoensis* (AAL12889); *Chiloscyllium punctatum* (AAL12884); *Monopterus albus* (XP_020470878); *Paralichthys olivaceus* (AIK29462); *Nothobranchius furzeri* (KAF7224727); *Oncorhynchus mykiss* (XP_021428825); *Danio rerio* (NP_571460); *Scobinancistrus pariolispos* (ADE34111).

**Figure 3 ijms-23-07546-f003:**
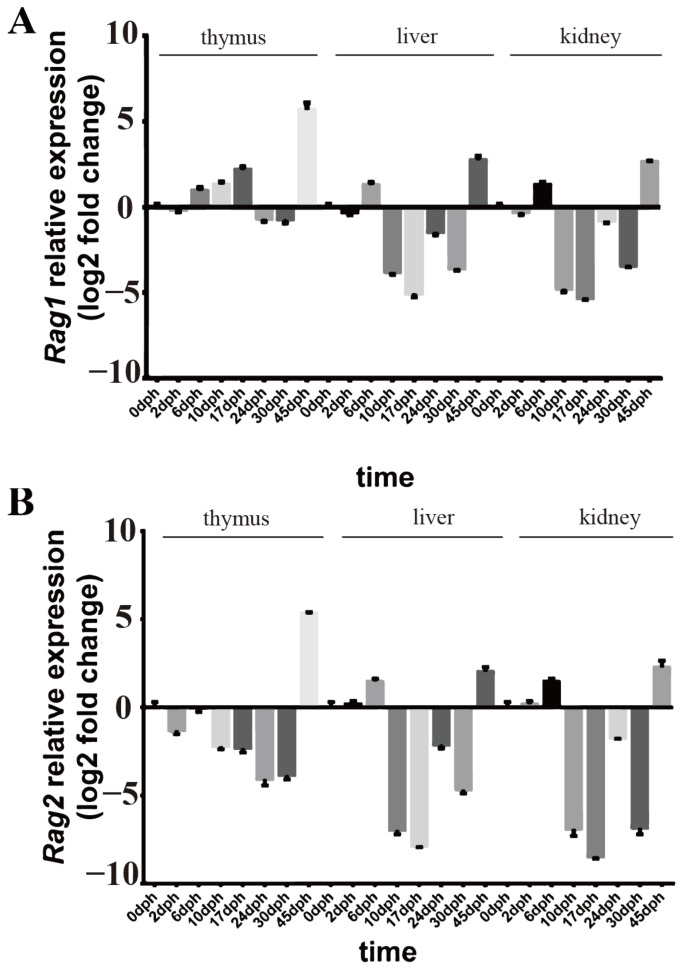
Real-time PCR determined relative mRNA expressions of *rag1* (**A**) and *rag2* (**B**) in the kidney, liver, and thymus from 0 to 45 dph of rice-field eel, with *ef1α* mRNA expression as the internal control. High expression is represented by a positive log2 expression ratio and low expression is represented by a negative log2 expression ratio; n = 3 independent experiments. Data are shown as the mean ± SD (n = 4).

**Figure 4 ijms-23-07546-f004:**
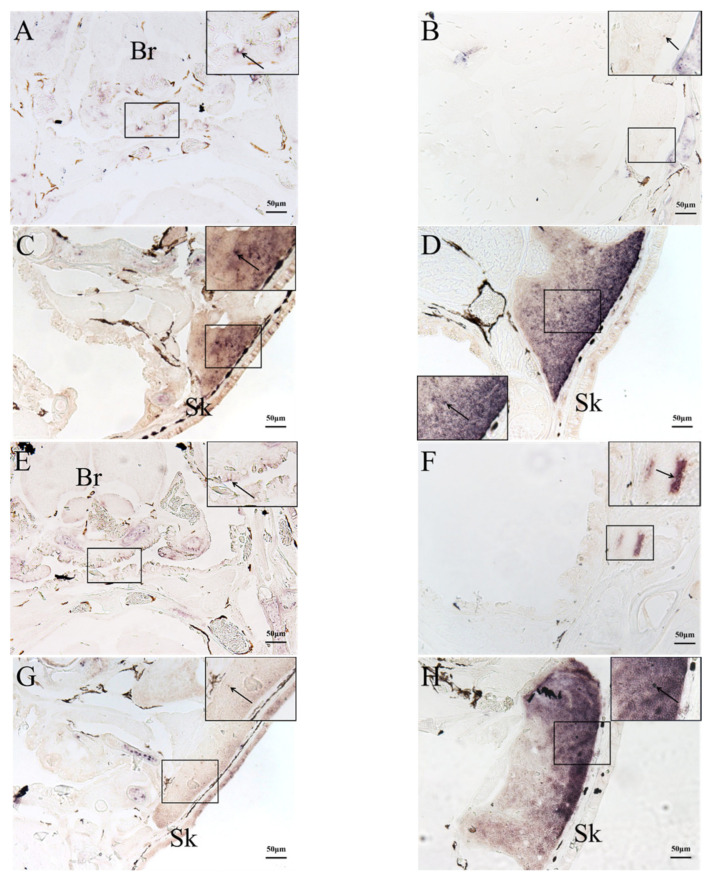
Expression of *rags* genes in thymus at different developmental stages. The expressions of (**A**) *rag1* and (**E**) *rag2* at 2 dph was observed at the thymic primordium epithelial layer. The expressions of (**B**) *rag1* and (**F**) *rag2* at 10 dph was observed mostly at the edge of the thymus. The expressions of (**C**) *rag1* and (**G**) *rag2* at 24 dph was scattered randomly throughout the entire thymus. The expressions of (**D**) *rag1* and (**H**) *rag2* at 45 dph was observed mostly in the sub-capsular region of the thymus, with little in the medulla. (**A**,**E**): 2 dph; (**B**,**F**): 10 dph; (**C**,**G**): 24 dph; (**D**,**H**): 45 dph. Bars: 50 µm. Arrows: *rags* signals. Br: brain. Sk: skin.

**Figure 5 ijms-23-07546-f005:**
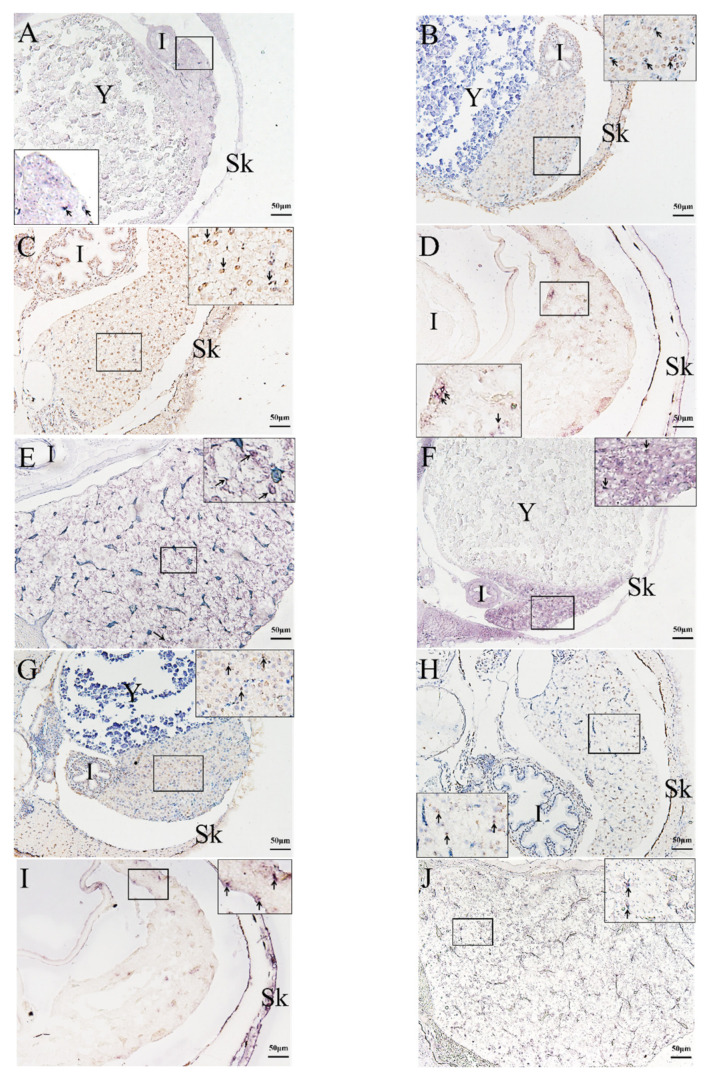
Expression of *rags* genes in the liver at different developmental stages. *Rag1* (**A**–**E**) and *rag2* (**F**–**J**) expression was randomly distributed in the entire liver. (**A**,**F**): 2 dph; (**B**,**G**): 6 dph; (**C**,**H**): 24 dph; (**D**,**I**): 30 dph; (**E**,**J**): 45 dph. (**B**,**C**,**G**,**H**): DAB staining, (**A**,**D**–**F**,**I**,**J**): NBT/BCIP staining. Bars: 50 µm. Arrow: *rags* signal. Y: yolk. I: intestine. Sk: skin.

**Figure 6 ijms-23-07546-f006:**
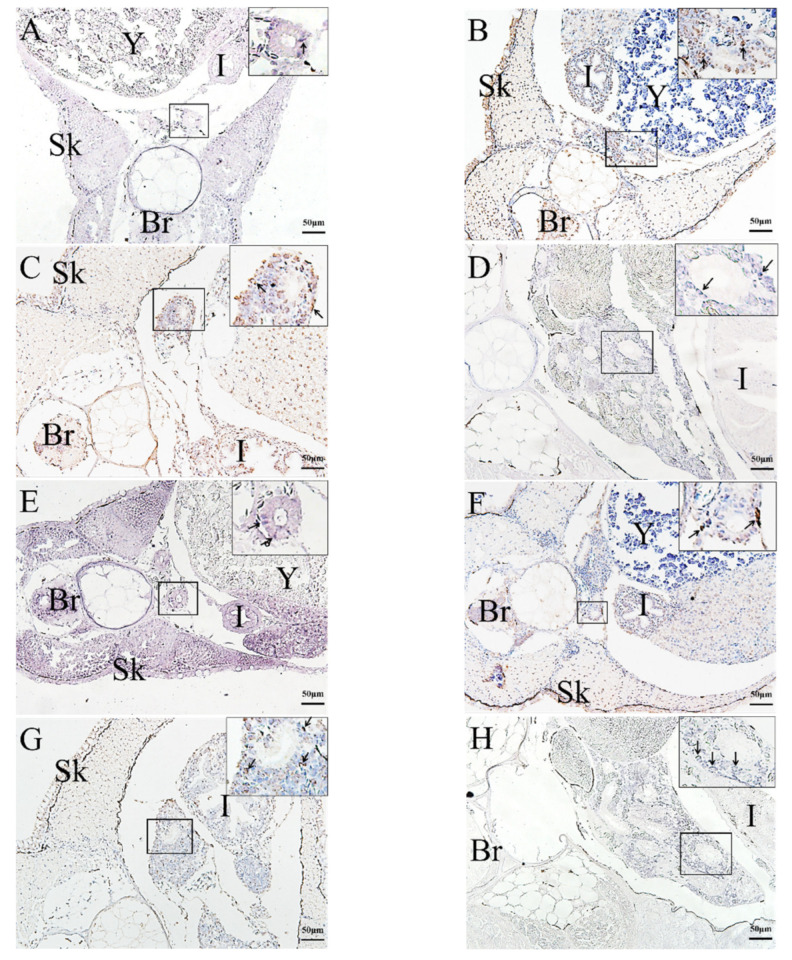
Expression of *rags* genes in the kidney at different developmental stages. *Rag1* (**A**–**D**) and *rag2* (**E**–**H**) expression was observed in the hematopoietic tissues of kidney. (**A**,**E**): 2 dph; (**B**,**F**): 6 dph; (**C**,**G**): 30 dph. (**D**,**H**): 45 dph. (**B**,**C**,**F**,**G**): DAB staining, (**A**,**D**,**E**,**H**): NBT/BCIP staining. Bars: 50 µm. Arrow: *rags* signal. Br: brain. Y: yolk. I: intestine. Sk: skin.

**Figure 7 ijms-23-07546-f007:**
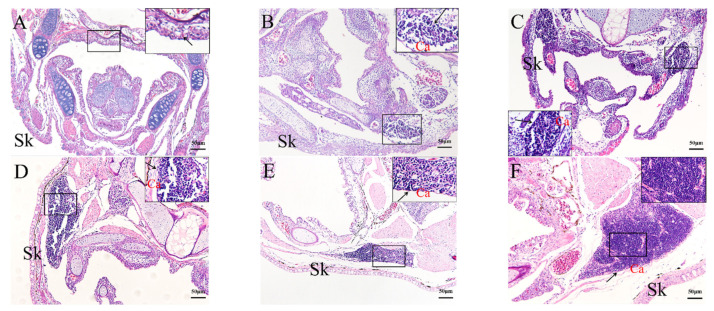
Hematoxylin and eosin staining of histological sections of the rice-field eel thymus from 2 dph to 45 dph. (**A**): 2 dph; (**B**): 6 dph; (**C**):10 dph; (**D**):17 dph; (**E**): 30 dph; (**F**): 45 dph. Bars: 50 µm. Black arrow: thymus. Ca: capsula. Sk: skin. Sections were stained with HE.

**Figure 8 ijms-23-07546-f008:**
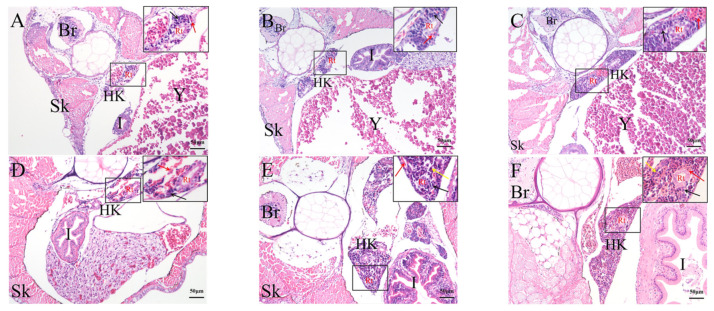
Hematoxylin and eosin staining of histological sections of the rice-field eel kidney from 2 dph to 45 dph. (**A**): 2 dph; (**B**): 6 dph; (**C**): 10 dph; (**D**): 17 dph; (**E**): 24 dph; (**F**): 45 dph. Bars: 50 µm. Hk: head kidney. Rt: renal tubule. Br: brain. Y: yolk. I: intestine. Sk: skin. Black arrow: lymphocytes. Red arrow: blood cells. Yellow arrow: granulocyte. Sections were stained with HE.

**Figure 9 ijms-23-07546-f009:**
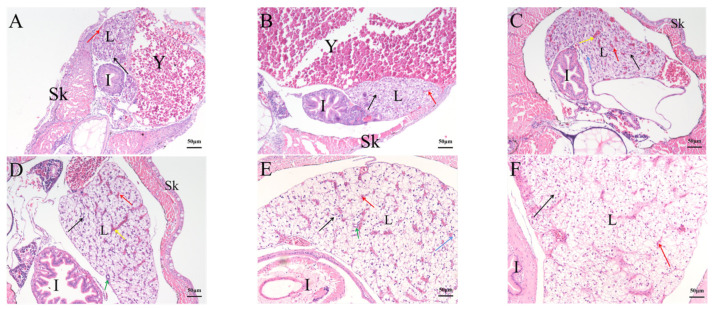
Hematoxylin and eosin staining of histological sections of the rice-field eel liver during development from 2 dph to 45 dph. (**A**): 2 dph; (**B**): 6 dph; (**C**): 17 dph; (**D**):24 dph; (**E**): 30 dph; (**F**): 45 dph. Bars: 50 µm. Black arrow: hepatic cord. Red arrow: hepatic sinusoid. Yellow arrow: central vein. Blue arrow: binuclear hepatocytes. Green arrow: bile canalicular. L: liver. Y: yolk. I: intestine. Sk: skin. Sections were stained with HE.

**Table 1 ijms-23-07546-t001:** Primers used in this study.

Gene Symbol	Primer Pairs Sequence (5′–3′)	Product Size (bp)	Accession Nos.
*rag1* (Real-time PCR)	F: TGCAGCTGATACCATCGGTC	128	XM020615225
	R: CTGAGGTGCAAGCACTGTCT		
*rag2* (Real-time PCR)	F: CTCATGCCCGACAGGAGTTT	114	XM020615222
	R: ACAGCAGGACAACGTAGTGG		
*ef1α* (Real-time PCR)	F: ATCCGTCGTGGATATGTGGC	125	XM020588923
	R: AGCACTGGGGCATAACCTTC		
*rag1-H* (Probe synthesis)	F: AAAGCCAAACTCAGAACTATCC	516	XM020615225
	R: TACACCTCGCCAATCTCATCT		
*rag2-H* (Probe synthesis)	F: CTTTACTTCGGTCGTGAGCCT	413	XM020615222
	R: GGGGACATCTTGATTGTTGGAG		

## Data Availability

The data that support the findings of this study are available from the corresponding author upon reasonable request.

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
