# Peer review of "Rag1 and Rag2 Gene Expressions Identify Lymphopoietic Tissues in Larvae of Rice-Field Eel (Monopterus albus)"

_ijms, 2022, doi:10.3390/ijms23147546_

Round 1

Reviewer 1 Report

The present manuscript by Liu and colleagues uses the Rag1/Rag2 expression to determine the pattern of lymphoid development in thymus kidney and liver in the rice-field eel (Monopterus albus), a teleost of economical relevance in China. Unfortunately, the manuscript provides little or no new information on teleost lymphoid development, an issue repeatedly studied in numerous fish species. Therefore, I don’t recommend the publication of manuscript in the present form and suggest to authors to take in consideration the following comments:

Introduction

. Authors would provide a brief description on the different role played by Rag1 and Rag 2 in the generation of T and B cell repertoire to justify the study of the two molecules.

. Authors emphasize that rice-field eel is “different from ordinary fish” (Line 38), could they to indicate those differences relevant for the biology of fish immune system?.

. Line 54. Not only head kidney, but also trunk/posterior kidney, are important lympho-hematopoietic organs in teleosts.

. Line 59. It has been reported that in some teleosts, particularly marine fish, liver and spleen appear earlier than thymus, but it is important to clarify that these organs do not contain lymphoid cells in these first stages of development. Indeed, teleost thymus is the first organ to become lymphoid.

Results/Discussion

. Fig 3. Why both liver and kidney present two peaks of Rag activity?. This issue would be discussed and correlated with the appearance of lymphoid cells in the distinct studied organs.

. In general, it is impossible to see the reported results in the provided figures; even inserts are too small to distinguish the described details, particularly in Figures 5 and 6.

. Rag 1 and Rag 2 expression shown in Figure 4A, E that authors describe associated to the “pharyngeal epithelium” does not correspond to the area occupied by the thymic primordium (see Figure 7). In any case, similar staining is seen in the muscles of oral cavity in the same Figure 4A, E.

. In the liver (Fig 5), the staining is minimal. Figures shown do not demonstrate conclusively presence of Rag transcripts in the liver.

. Line 167. It is impossible to detect the presence of Hassall corpuscles in the indicated Figure. On the other hand, it is assumed that Hassall bodies do not exist in teleost thymus.

. Line 212. Some relevant references on teleost thymus would be discussed (Willet et al 1999 Dev Dyn 214, 323; Zapata 2022 Biology 11, 747).

. Lines 215-219. To discuss the obtained results with those reported on the lymphoid organs of Xenopus and Urodeles is, in my opinion, inadequate because the immune system of teleosts and amphibians is very different. Even the condition of the lymphoid organs of Xenopus is special and does not occur in other more advanced Anurans, such as frogs and toads.

. Line 251-252. The author comments on the lymphopoietic capacities of teleost kidney are wrong, lympho-haematopoietic tissue occurs between the renal tubules in all studied teleost species.

. Page 11. The description of liver development is absolutely irrelevant for the article aim and would be avoided.

. Line 310. As above indicated, in all teleosts kidney develops earlier than thymus, but this last organ is the first one for becoming lymphopoietic.

Author Response

Response to Reviewer’s Comments

Dear editor,

Thank you for your letter and for the reviewer’s valuable comments and suggestions concerning our manuscript entitled ‘Rag1 and rag2 gene expressions identify lymphopoietic tissues in larvae of rice-field eel (Monopterus albus)’ (Manuscript ID: ijms-1762706). According to the comments, we have made all the necessary changes in the revised manuscript. All the revisions in the manuscript and the supporting information have been highlighted in red. In addition, the point-to-point responses to the reviewer’s comments are also provided.

Best regards,

Yours sincerely,

Yuding Fan

Reviewer #1:

The present manuscript by Liu and colleagues uses the Rag1/Rag2 expression to determine the pattern of lymphoid development in thymus kidney and liver in the rice-field eel (Monopterus albus), a teleost of economical relevance in China. Unfortunately, the manuscript provides little or no new information on teleost lymphoid development, an issue repeatedly studied in numerous fish species. Therefore, I don’ t recommend the publication of manuscript in the present form and suggest to authors to take in consideration the following comments:

Introduction:

  1. Authors would provide a brief description on the different role played by Rag1 and Rag2 in the generation of T and B cell repertoire to justify the study of the two molecules.

Response: Thank you for this advice. We have added these information in the revised manuscript (Line 70-75). The details are as follows:

Ig is secreted by B lymphocytes, while TCR is expressed by T lymphocytes. The recombination signal sequence exists in the V (D) J region of Ig and TCR. Rag1 can recognize and bind RSS alone, while rag2 can bind to DNA only in the presence of rag1. In addition, rag1 and rag2 genes can form complexes, which have higher specificity and stability, and can be recognized more effectively [18, 19].

  1. Authors emphasize that rice-field eel is “different from ordinary fish” (Line 38), could they to indicate those differences relevant for the biology of fish immune system?

Response: We understand the reviewer’s concern. In fact, “the difference” we mean the different physiological characteristics between the rice-field eel and other fishes, for example, a sex reversal phenomenon, and it was irrelevant for the biology of fish immune system. The difference relevant for the immune system need to be discussed in the following chapters of this article.

  1. Line 54. Not only head kidney, but also trunk/posterior kidney, are important lympho-hematopoietic organs in teleosts.

Response: Thank you for this advice. According to your advice, we have changed this description in the revised manuscript (Line 52-54). The details are as follows:

Compared with mammals, the rice-field eel, as a kind of bony fish, lacks bone marrow and lymph nodes [8], and its lymphopoietic tissues mainly comprise the thymus, kidney, and spleen [9].

  1. Line 59. It has been reported that in some teleosts, particularly marine fish, liver and spleen appear earlier than thymus, but it is important to clarify that these organs do not contain lymphoid cells in these first stages of development. Indeed, teleost thymus is the first organ to become lymphoid.

Response: Thank you for this advice. According to the suggestion, we have added relevant description in the revised manuscript (Line 57-61). The details as follows:

Although the earliest lymphoid organ in freshwater fish is the thymus and the kidney and spleen developed earlier than thymus in some seawater fish, the thymus is also the first organ to become lymphoid [10]. In rock bream [11] and Atlantic cod [12], the earliest immune organ becoming lymphoid is the kidney.

Results/Discussion:

  1. Fig 3. Why both liver and kidney present two peaks of Rag activity?. This issue would be discussed and correlated with the appearance of lymphoid cells in the distinct studied organs.

Response: Thank you for this advice. We have added these information in the revised manuscript (Line 234-239). The details are as follows:

The activity of rags induces V (D) J rearrangement during B cells development. In B cell development, rags transcriptions levels were higher in pro-B cells and pre-B cells and lower in lymphoid stem cells, immature B cells, naive B cells and mature B cells [32]. In this study, the presence of two peaks of rags transcriptions might indicate the presence of pro-B cells and pre-B cells in the liver and kidney. Therefore, the liver and kidney were considered to be lymphopoiesis tissues.

  1. In general, it is impossible to see the reported results in the provided figures; even inserts are too small to distinguish the described details, particularly in Figures 5 and 6.

Response: Thank you for this advice. We have readjusted the Figures 5 and 6, the signals were indicated by arrows.

  1. Rag1 and Rag2 expression shown in Figure 4A, E that authors describe associated to the “pharyngeal epithelium” does not correspond to the area occupied by the thymic primordium (see Figure 7). In any case, similar staining is seen in the muscles of oral cavity in the same Figure 4A, E.

Response: Thank you for this advice, we have changed a new Figure 7A.

  1. In the liver (Fig 5), the staining is minimal. Figures shown do not demonstrate conclusively presence of Rag transcripts in the liver.

Response: Thank you for this advice. We have readjusted the Figures 5, the signals were indicated by arrows.

  1. Line 167. It is impossible to detect the presence of Hassall corpuscles in the indicated Figure. On the other hand, it is assumed that Hassall bodies do not exist in teleost thymus.

Response: Thank you to this advice, we have revised the term in the revised manuscript (Line 173-176).

  1. Line 212. Some relevant references on teleost thymus would be discussed (Willet et al 1999 Dev Dyn 214, 323; Zapata 2022 Biology 11, 747).

Response: Thank you to this advice. According to your advice, we have added some relevant discussion in the revised manuscript (Line 220-224). The details are as follows:

 The thymus is an important immune organ. Previous studies showed the relevant characteristics of the thymus of teleost fish, for example, the main lymphoid and hematopoietic tissue development of the zebrafish [27], and the origins of T and B lymphopoiesis in agnatha and jawed fish [28].

  1. Lines 215-219. To discuss the obtained results with those reported on the lymphoid organs of Xenopus and Urodeles is, in my opinion, inadequate because the immune system of teleosts and amphibians is very different. Even the condition of the lymphoid organs of Xenopus is special and does not occur in other more advanced Anurans, such as frogs and toads.

Response: Thank you for your suggestion. We have added the information more discussion about this in the revised manuscript (Lines 227-231). The details are as follows:

Rag1 gene expressed at the surface of thymus, and rag2 gene expressed in both thymus and head kidney, which indicated that thymus and head kidney was primitive lymphoid tissue in rainbow trout [16, 23]. Rag1 and rag2 genes were expressed in the head and body kidneys of Paralichthys olivaceus, therefore, the kidney was the lymphoid organ of Paralichthys olivaceus [29].

  1. Line 251-252. The author comments on the lymphopoietic capacities of teleost kidney are wrong, lympho-haematopoietic tissue occurs between the renal tubules in all studied teleost species.

Response: Thank you for your suggestion. We have revised this sentence to “The kidney of teleost fish is usually divided into head, middle, and posterior kidneys. The middle kidney and posterior kidney are not only urinary organs but also hematopoietic tissue, which function urination, regulation of osmotic pressure and hemopoiesis.”(Lines 271-273).

  1. Page 11. The description of liver development is absolutely irrelevant for the article aim and would be avoided.

Response: Thank you for your suggestion. We have deleted the irrelevant description in the revised manuscript (Lines 298-317).

  1. Line 310. As above indicated, in all teleosts kidney develops earlier than thymus, but this last organ is the first one for becoming lymphopoietic.

Response: Thank you for your suggestion. We have changed this description in the revised manuscript (Lines 328-330). The details as follows:

Accordingly, in the rice-field eel, the liver and the kidney developed earlier than the thymus, and the kidney was the first immune organ becoming lymphoid, which is similar to the development order of rock bream [11] and Atlantic cod [12].

Reviewer 2 Report

This is an interesting work. The expression of rag1 and rag2 was studied on ricefield eel on three different tissued to evaluate temporal and spatial expression patterns in different development stages. The tables and the graphs are appropriate and the text is clear.

The references all over the paper are written according to Oxidative Medicine and Cellular Longevity rules, but the references at the end of the article should be revised.

Specific comments

Introduction

Line 29- review tipping errors, such as “TThe rice-field…”, italics in the specie name….

Results

Line 113: Please, include the P values in the text. You can then describe if the gene expression variation is significant in each comparison when it is done. For example, in line 117, when you described: “…the expression level of rag1 showed a roughly increasing trend from 0 dph to 17 dph, after which it decreased, and maintained at a lower level from 24 dph to 30 dph…”This could be a little confusing because you said the rag1 expression is decreased and maintained at lower level from 24 dph to 30 dph (although no differences were observed with respect to 0dph). In the case of rag2 expression, you can speak of increases or decrease because they are significant. For avoid any confusion, please specify if the differences are significant in any case.

Line 159: The histological analysis results showed subjective terms such as: many, a few… Maybe could be interesting to make measurements of the micrographs by image analyse tools, such as Image J, etc.

Discussion

No data available about thymus or head kidney development on Paralichthys olivaceus? A comparison could be interesting.

Line 205, 246, 273, 285, 288, 296 and 315: italics in the specie name

Line 206: as well is true that thymus, liver and kidney have the same significant increase of rag1 and rag 2 expression, the expressions of rag1 and rag2 genes in rice-field eel didn`t be similar. Please, change the way to explain these observations to avoid any confusion about that.

Line 252: Please, introduce a reference.

Material and methods

Line 324: Please include the body length, body weight, …

Line 325: Please, include the water conditions (filtered, recirculate, volume, light cycle, …).

Line 326: Please include feed procedures (times per day, kind of food, …).

Line 327: Animals sacrificed for any experimental time was 15. How was the aquaria disposition? All the animals were in the same aquaria? Please, clarify the technical an biological replicates.

Line 329: The phylogenetic tree analysis methodology is too short. It should be better explained. For example, specify the method that you have used in Clustal W software. Moreover, 1000 bootstraps are too few for this kind of analysis. In bibliography, papers normally describe 10000 bootstraps. This analysis could take a few hours and 0.05 of evolutive distance is ok, but please, I suggest that justify the 1000 bootstraps.

Line 331: I suppose that you want to say:  -80ºC. Please correct.

Line 431: Please, clarify in the figure legends.

Line 343-351: The endogenous gene (housekeeping) is ef1a. Another housekeeping is desirable. Is it possible? Nowadays, the journals are looking for 2-3 housekeeping. You justify this because your previous publication. Please, introduce another gene, or reference other works for justify your procedure.

Author Response

Response to Reviewer’s Comments

Dear editor,

Thank you for your letter and for the reviewer’s valuable comments and suggestions concerning our manuscript entitled ‘Rag1 and rag2 gene expressions identify lymphopoietic tissues in larvae of rice-field eel (Monopterus albus)’ (Manuscript ID: ijms-1762706). According to the comments, we have made all the necessary changes in the revised manuscript. All the revisions in the manuscript and the supporting information have been highlighted in red. In addition, the point-to-point responses to the reviewer’s comments are also provided.

Best regards,

Yours sincerely,

Yuding Fan

Reviewer #2:

This is an interesting work. The expression of rag1 and rag2 was studied on rice-field eel on three different tissued to evaluate temporal and spatial expression patterns in different development stages. The tables and the graphs are appropriate and the text is clear.

The references all over the paper are written according to Oxidative Medicine and Cellular Longevity rules, but the references at the end of the article should be revised.

Response: We have revised the references.

Specific comments:

Introduction:

  1. Line 29- review tipping errors, such as “TThe rice-field…”, italics in the specie name....

Response: Thank you for this advice. According to your advice, we have changed the specie name in the revised manuscript (Line 29). The details are as follows:

The rice-field eel (Monopterus albus), belonging to the family Synbranchidae, is an important economic fish species in China [1].

Results:

  1. Line 113: Please, include the P values in the text. You can then describe if the gene expression variation is significant in each comparison when it is done. For example, in line 117, when you described: “…the expression level of rag1 showed a roughly increasing trend from 0 dph to 17 dph, after which it decreased, and maintained at a lower level from 24 dph to 30 dph…”This could be a little confusing because you said the rag1 expression is decreased and maintained at lower level from 24 dph to 30 dph (although no differences were observed with respect to 0dph). In the case of rag2 expression, you can speak of increases or decrease because they are significant. For avoid any confusion, please specify if the differences are significant in any case.

Response: According to the review’s suggestion, we have changed this description in the revised manuscript (Line 123-127). The details are as follows:

According to the qRT-PCR results, significant expression (p ≤ 0.05) of rag1 and rag2 was detected in the kidney, liver, and thymus from 0 to 45 dph in rice-field eel (Figure 3A, B). In the early development stage of the thymus, the expression level of rag1 showed a roughly increasing trend from 0 dph to 17 dph, and it had no significant change from 24 dph to 30 dph.

  1. Line 159: The histological analysis results showed subjective terms such as: many, a few… Maybe could be interesting to make measurements of the micrographs by image analyse tools, such as Image J, etc.

Response: Thank you for this advice. We have deleted these subjective terms.

Results:

  1. No data available about thymus or head kidney development on Paralichthys olivaceus? A comparison could be interesting.

Response: Thank you for this advice. According to the suggestion, we have added more discussion about thymus or head kidney development on Paralichthys olivaceus (Lines 230-231, 256-259, 283-284). The details as follows:

Rag1 and rag2 genes were expressed in the head and body kidneys of Paralichthys olivaceus, therefore, the kidney was the lymphoid organ of Paralichthys olivaceus [29] .

The thymic primordia of carp appeared about 2 dph [36], and that of sea bream ap-peared at 11 dph [37], that of Paralichthys olivaceus appeared at 10 ~ 15 dph [38], while the thymic primordia of Atlantic sturgeon appeared at 48 dph [39].

The kidney of Paralichthys olivaceus appears at the time of hatching stage [38].

  1. Line 205, 246, 273, 285, 288, 296 and 315: italics in the specie name.

Response: Thank you for this advice. According to your advice, we have changed these specie name in the revised manuscript (Line 212, 267, 296, 306, 313 and 334). The details are as follows:

Herein, we observed that the rice-field eel RAG1 and RAG2 proteins were phylogenetically closely related to their homologs in Paralichthys olivaceus.

However, the thymus of the rice-field eel did not show lobulation, which was different from that of Epinephelus obliquus [41] and Chinese sturgeon [42].

While in Paralichthys olivaceus and golden head snapper, ...

The liver rudiment of Epinephelus obliquus appeared about 3 dph, ...

There are three arrangements of hepatocytes in the liver, one of which, such as in Culter Alburnus, ...

In conclusion, rice-field eel RAG1 and RAG2 are closely related to their homologs in Paralichthys olivaceus.

  1. Line 206: as well is true that thymus, liver and kidney have the same significant increase of rag1 and rag2 expression, the expressions of rag1 and rag2 genes in rice-field eel didn`t be similar. Please, change the way to explain these observations to avoid any confusion about that.

Response: Thank you for this advice, and we have changed the description in the revised manuscript (Lines 212-216). The details are as follows:

Rag1 and rag2 genes expression of the liver and kidney in rice-field eel were similar, starting as early as 0 dph and peaking at 45 dph. However, in the thymus of rice-field eel, rag1 increased significantly from 0 dph to 17 dph, and peaking at 45dph, while rag2 decreased from 0 to 30 dph and then increased at 45 dph.

  1. Line 252: Please, introduce a reference.

Response: Thank you for this advice. The reference has been added in the revised manuscript (Lines 273).

Material and methods:

  1. Line 324: Please include the body length, body weight, ….

Response: Thank you for this advice. Unfortunately, we didn't record data on body length and weight. However, we sampled fish of similar size in each time period according to the time sequence.

  1. Line 325: Please, include the water conditions (filtered, recirculate, volume, light cycle, …).

Response: According to the reviewer’s suggestion, we have added information related to the water conditions (Lines 345-347). The details are as follows:

 They were distributed in a 30 x 100 cm boxes with shading, and the water was changed once a day. The aerator was used to provide the oxygen needed by the rice-field eel..

  1. Line 326: Please include feed procedures (times per day, kind of food, …).

Response:Thank you for this advice. We have added feed procedures in the revised manuscript (Lines 347). The details are as follows:

 Fish were fed with red worm once a day.

  1. Line 327: Animals sacrificed for any experimental time was 15. How was the aquaria disposition? All the animals were in the same aquaria? Please, clarify the technical and biological replicates.

Response: Thank you for this advice. The specific aquaria disposition is described above, and we have added the information about the animals in the revised manuscript (Lines 348-352). The details are as follows:

120 fishes spawned at one time were used as an experiment. Every 40 fishes were kept in the same tank, for a total of 3 tanks. Five fishes were taken from each tank at each time point, and a total of fifteen fishes were used as a sample. We used three different batches of fish as biological replicates. The experiments were performed in triplicate as technical replicates.

  1. Line 329: The phylogenetic tree analysis methodology is too short. It should be better explained. For example, specify the method that you have used in Clustal W software. Moreover, 1000 bootstraps are too few for this kind of analysis. In bibliography, papers normally describe 10000 bootstraps. This analysis could take a few hours and 0.05 of evolutive distance is ok, but please, I suggest that justify the 1000 bootstraps.

Response: Thank you for this advice. 1000 bootstraps have been used by researchers many times, for example, [31] Jiang et al 2018 Dev Comp Immunol 87 24-35, and Shen et al 2019 Freshwater Fisheries 49(4) 16-21. Therefore, we also used 1000 bootstraps to make a phylogenetic tree analysis.

  1. Line 331: I suppose that you want to say: -80º Please correct.

Response: Thank you for this suggestion. We have corrected the symbols in the revised manuscript (Lines 355-357).

  1. Line 431: Please, clarify in the figure legends.

Response: According to the reviewer’s suggestion, we have added information to the revised manuscript (Lines 104-106, 114-116). The details as follows:

The phylogenetic tree was constructed based on the full-length aminoacid sequences by the maximum likelihood (ML) method and the bootstrap value was set at 1000.

  1. Line 343-351: The endogenous gene (housekeeping) is ef1a. Another housekeeping is desirable. Is it possible? Nowadays, the journals are looking for 2-3 housekeeping. You justify this because your previous publication. Please, introduce another gene, or reference other works for justify your procedure.

Response: Thank you for this advice. Although, many journals are looking for 2-3 housekeeping, but one suitable reference gene is also possible. Many researchers used one reference gene to analysis gene expression. For example, [31] Jiang et al 2018 Dev Comp Immunol 87 24-35, and Ding et al Acta Agriculture Boreali-Sinica 2012 27 6-11.